# Syntheses and reactivities of strained fused-ring metallaaromatics containing planar eleven-carbon chains

Binbin Xu[1,3], Wei Mao[1,3], Zhengyu Lu[1,3], Yuanting Cai[1], Dafa Chen[1] ✉ & Haiping Xia [1,2] ✉

Carbolong complexes are one of the primary types of metallaaromatics, and they include metallapentalynes and metallapentalenes. A series of 7C-10C and 12C-carbolong complexes with planar ligand skeletons respectively containing 7-10 and 12 carbon atoms in their backbones, have been previously reported. Herein, two classes of strained substances, metallabenzyne-fused metallapentalenes and metallabenzene-fused metallapentalynes, were prepared, both representing 11C-carbolong complexes with a planar carbon-chain ligand. Furthermore, the former type is also the carbolong derivatives containing a metallabenzyne skeleton, another primary metallaaromatic framework. Metallabenzyne-fused metallapentalenes show versatile reactivities, and the most interesting one is the metal carbyne bond shift from a 6-membered to a more strained 5-membered ring, affording the above-mentioned metallabenzene-fused metallapentalyne. This work makes carbolong chemistry more complete, and provides a method to achieve metallabenzynes, which is anticipated to concurrently advance the development of these two types of metallaaromatics.

Aromatic compounds are one of the most important species in chemistry. It is estimated that about half of the registered chemicals ( > 30 million) are aromatic compounds[1]. Most aromatics are organic compounds, and their aromaticity is derived from $\pi$-conjugation of $p$-orbitals. In comparison, metallaaromatics in which at least one transition metal is involved in the aromatic ring, are a special kind of aromatic complexes involving $d_\pi$-$p_\pi$ conjugation[2–13]. Although it is still not comparable with organic aromatic compounds in terms of quantity, much progress has been made on the research of metallaaromatic complexes, since several hypothetical metallabenzenes were theoretically predicted to be aromatic by Thorn and Hoffmann in 1979[14] and the first real metallabenzene was experimentally characterized by Roper and his co-workers in 1982[15]. After more than four decades of development, most metallaaromatic complexes can be classified into six categories, namely metallabenzenes[16–23], metallabenzynes[24–27], heterometallaaromatics[28–32], dianion metalloles[33,34], spiro metalloles[35–37], and carbolong complexes[38–47].

Carbolong complexes, which were defined as metal bridgehead polycyclic frameworks featuring a carbon chain with at least seven carbon atoms ( ≥ 7 C) coordinated to a metal atom via not less than three metal–carbon σ bonds, are a class of metallaaromatics with two fused five-membered metallacycles as the basic unit[7,38–53]. The first discovered carbolong complexes were metallapentalynes reported in 2013 by our group, and because of containing a planar seven-carbon chain in the backbone, they are also called as 7C-carbolong complexes[48]. Since then, a series of conjugated planar 8C-10C[49–52] and 12C-carbolong[53] complexes have been developed. These complexes have the properties of both aromatic and organometallic compounds, and often exhibit broad absorption spectra, as a result, they have been applied in several areas, such as solar cells, biomedical

[1]Shenzhen Grubbs Institute and Guangdong Provincial Key Laboratory of Catalysis, Department of Chemistry, Southern University of Science and Technology, Shenzhen, China. [2]Southern University of Science and Technology Guangming Advanced Research Institute, Shenzhen, China. [3]These authors contributed equally: Binbin Xu, Wei Mao, Zhengyu Lu. ✉e-mail: chendf@sustech.edu.cn; xiahp@sustech.edu.cn

area and catalysis[54]. However, one notable omission is that 11C-carbolong complexes with a planar carbon-chain ligand have not been synthesized so far, although a non-planar one with one of the two M–C bonds in the six-membered ring locating vertical to the metallapentalene plane was reported[55]. Besides, the other issue that needs to be mentioned is that the interesting and important M≡C bond, which exhibits rich reactivities, has not yet appeared in a system larger than 7C-carbolong framework (metallapentalynes)[48,50,55].

In order to make the family of carbolong complexes a more complete whole, we have been committed to prepare planar 11C-carbolong skeletons for years. Herein, we present the syntheses and reactivities of several metallabenzyne-fused metallapentalenes (**I** in Fig. 1), which are not only 11C-carbolong complexes with a planar carbon-chain ligand, but also carbolong complexes that contain a metallabenzyne framework, another type of metallaaromatics. Furthermore, they are also carbolong complexes containing a M≡C bond that is not within the five-membered ring. Due to the presence of a strained metallacycle, these complexes show versatile reactivities, and a series of derivatives, including substances such as one metallabenzene-fused metallapentalyne (**II**), one dimetallic complex, and one 12C-carbolong complex, were obtained (Fig. 1). The metallabenzene-fused metallapentalyne is also an 11C-carbolong complex with a flat carbon skeleton, and its formation underwent a M≡C bond shift from metallabenzyne ring to a more strained metallapentalene ring, which is different from those migrations between two five-membered rings in metallapentalynes[45,48,56]. Interestingly, the M≡C bond in this metallabenzene-fused metallapentalyne could shift back to the six-membered ring upon the addition of PhC≡CLi, affording another metallabenzyne-fused metallapentalene. The obtained products show broad absorption bands in the UV-Vis−NIR region, especially the 12C-carbolong complex, whose absorption extends to over 1000 nm. Notably, the absorption of this 12C-carbolong complex is also broader than the previously discovered 12C-carbolong complexes[53], which is because, in this case, all the twelve carbon atoms in the backbone are sp²-hybridized, making this skeleton a fully conjugated system. Moreover, the photothermal performance of one of these complexes has been tested and satisfactory results were gained, demonstrating their potential to be photofunctional materials.

## Results

### Synthesis of metallabenzyne-fused metallapentalenes through the reduction of metallabenzooxirene-fused metallapentalenes

Propargylic alcohols are good 3 C synthons[57], therefore, we first tried the reactions of the 8C-carbolong complex **1**[49] with propargylic alcohols, to see the possibility of constructing 11C-carbolong complexes. When complex **1** was treated with 3-phenylprop-2-yn-1-ol (**2a**) in CH₂Cl₂ at room temperature (rt) for 24 hours under air, complex **3a** was isolated in 59% yield (Fig. 2a).

The X-ray crystal structure of **3a** is shown in Fig. 2b (left). As we expected, it is an 11C-carbolong derivative, and contains an extra fused

osmaoxirane ring, therefore, **3a** can also be regarded as an osmabenzooxirene-fused osmapentalene. The four fused rings have good planarity, with a mean deviation of 0.061 Å from the least-squares plane. The ³¹P{¹H} NMR spectrum exhibits two signals at 12.55 and −9.39 ppm, attributed to the phosphonium group and PPh₃ ligands, respectively, consistent with its planar skeleton. The chemical shift of C1*H* proton is observed at 12.29 ppm, near to those of the *ortho*-protons of osmapentalenes[38–53].

Thereafter, in order to verify the universality of the reaction, several other propargylic alcohols **2b-e** and propargyl phenyl ether (**2f**) were mixed with complex **1** in the presence of AgBF₄, respectively, and complexes **3b-f** possessing structures similar to that of **3a** were afforded in the yields of 42-70% (Fig. 2a). These products were characterized by NMR spectroscopy and HRMS spectrometry, and furthermore, **3c** was analyzed by X-ray single crystal diffraction (Supplementary Fig. 81). What should be mentioned is that complex **3f** does not contain the phenyl group of **2f**, as evidenced by the molecular ion peak of 1133.2814 (calculated m/z = 1133.2840), and the ¹H NMR for the C9*H* at 7.53 ppm.

The oxygen atom in the osmaoxirane ring might come from the 3 C synthons, O₂, or trace of H₂O in the system. To figure out the origin, ¹⁸O-labeled **2a** (¹⁸O-**2a**)[58] was synthesized and added to a mixture of **1** and AgBF₄ in CH₂Cl₂, resulting in the production of ¹⁸O-**3a** (Fig. 2a), as demonstrated by the HRMS spectrum (Supplementary Fig. 78), and the addition of excess H₂O did not influence the results. Therefore, it is clear that the oxygen atom is from propargylic alcohols when they were used as the reactants. Based on this result, a proposed mechanism of **1** with propargylic alcohols is shown in Supplementary Fig. 2.

To further determine the situation of propargyl phenyl ether (**2f**), the reaction of this substance with **1** and AgBF₄ in the presence of excess H₂¹⁸O was conducted, and the product was also labeled by ¹⁸O (¹⁸O-**3f**) in the yield of 29% (Supplementary Fig. 1), as evidenced by HRMS (Supplementary Fig. 79). We also tried the reaction of **2f** with **1** and AgBF₄ in extra-dry CH₂Cl₂ (the concentration of H₂O is 50 ppm) under a N₂ atmosphere, however, only trace of **3f** was detected. Therefore, it was H₂O from solvent and air that attacked the skeleton of **2f** and caused cleavage of the PhO–C bond during the reaction, which is not unexpected considering that the Ar–O bond is typically more inert than alkyl–O bond in aryl alkyl ethers.

Previously, there was a report that metallabenzooxirenes could be synthesized by oxidation of metallabenzynes[59]. Although the reverse reactions have not been discovered, we were curious if our metallabenzooxirene-fused metallapentalenes, each featuring a metallabenzooxirene moiety, could be reduced to metallabenzyne-fused metallapentalenes. Consequently, complexes **3a** and **3b** were respectively treated with LiAlH₄, followed by the addition of NaBPh₄ or NaCl for anion exchange, leading to the formation of complexes **4a** (71% yield), **4b-BPh₄** (75% yield) and **4b-Cl** (70% yield) (Fig. 2a).

Complexes **4a**, **4b-BPh₄** and **4b-Cl** have similar structures, and only **4b-Cl** is selected for discussion. As displayed in Fig. 2 (right), **4b-Cl** comprises a planar [5,5,6]-fused tricyclic framework (the mean deviation from the least-squares plane consisting Os1 and C1–C11 is 0.041 Å). The C–C bond lengths within the fused system range from 1.356(3) to 1.450(2) Å, and the Os1–C1, Os1–C4 and Os1–C7 lie in the range of 2.0809(17)−2.1342(17) Å, all comparable to the corresponding bonds in complex **3a** (C–C: 1.357(6)−1.442(6) Å; Os–C: 2.001(4)−2.110(4) Å). In contrast, the Os1–C11 bond is significantly shortened to 1.8503(18) Å, which is in the range of the Os≡C distances in osmabenzynes[24–27,60]. The carbyne carbon bond angle is 146.65(15)°, similar to those of Jia's osmabenzynes (148.3(6)−154.9(9)°) as well[12]. Therefore, complex **4b-Cl** can be considered as an osmabenzyne-fused osmapentalene. Similar to osmabenzynes, **4b-Cl** should also have other resonance structures (Supplementary Fig. 3)[12,60].

The NMR spectra of **4b-Cl** are consistent with its X-ray structure, and suggest it is an aromatic complex. The C1*H* is located at a

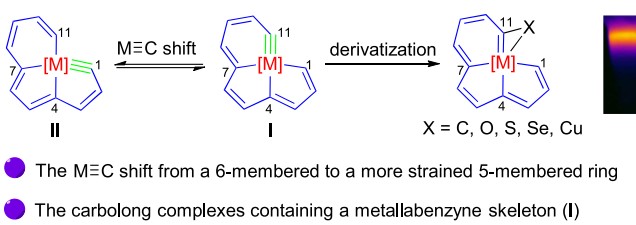

The M≡C shift from a 6-membered to a more strained 5-membered ring

The carbolong complexes containing a metallabenzyne skeleton (**I**)

The planar 11C-carbolong skeletons (**I** and **II**)

Broad UV-Vis−NIR absorptions and good photothermal property

**Fig. 1 | Main research content.** Structure and reactivities of metallabenzyne-fused metallapentalene (**I**), infrared thermal image of one of the products, and the main highlights of this work.

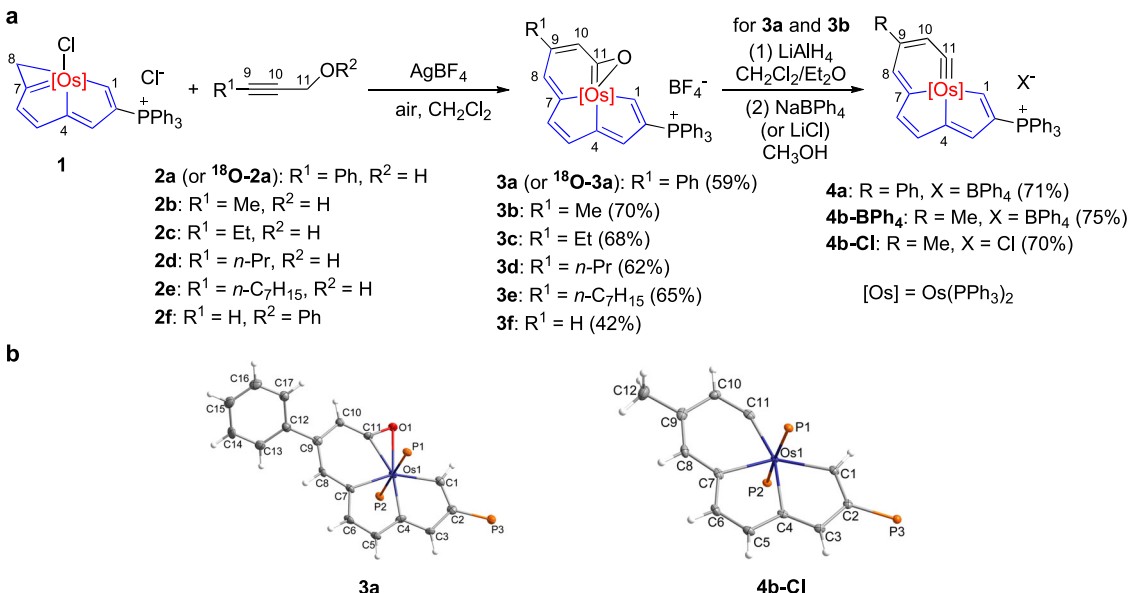

**Fig. 2 | Synthesis and structures of metallabenzooxirene-fused metallapenta-lenes and metallabenzyne-fused metallapentalenes. a** Synthesis of metallabenzooxirene-fused metallapentalenes and metallabenzyne-fused metallapentalenes. **b** Structures for the cations of complexes **3a** (left) and **4b-Cl** (right) with thermal ellipsoids drawn at the 50% probability level (phenyl groups in PPh₃ have been omitted for clarity).

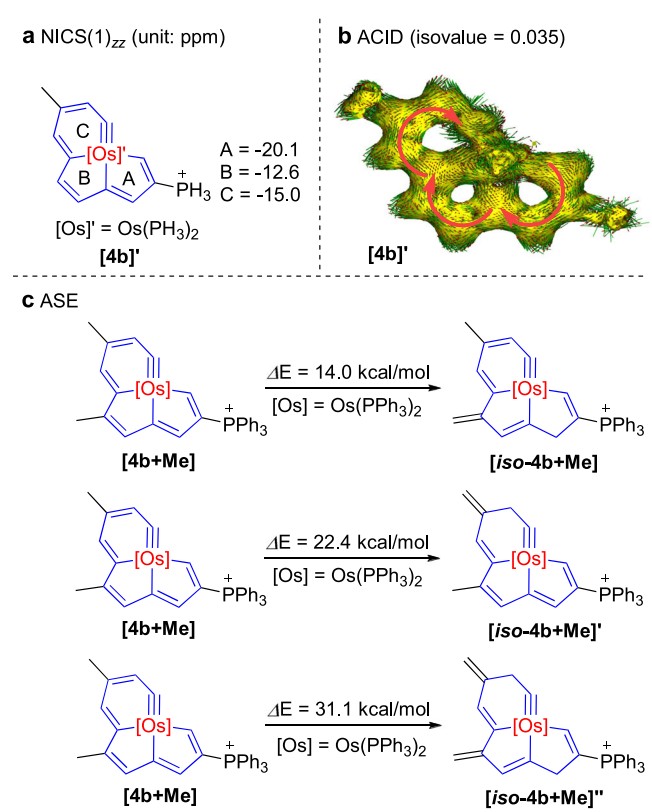

**Fig. 3 | Evaluation of aromaticity for complex 4b-Cl by DFT calculations.**
**a** NICS(1)$_{zz}$ values of the model complex **[4b]′**. **b** ACID plot of **[4b]′** with an iso-surface value of 0.035; the magnetic field vector is orthogonal to the ring plane and points upward (aromatic species exhibit clockwise diatropic circulations). **c** ASE values calculated on the model complex **[4b + Me]**.

osmabenzynes (3.77–4.26 ppm)[61,62]. The $^{13}C\{^1H\}$ NMR spectrum shows that the C1, C4, and C7 atoms resonate at 212.27, 201.70, and 219.65 ppm, respectively, close to those of metal-bound carbons in **3a** (210.96–191.92 ppm). On the other hand, the C11 signal is at 334.77 ppm, which is a characteristic position for Os≡C atoms in osma-benzynes (> 277.9 ppm)[59–62], and due to the coupling of the two Os-$PPh_3$ ligands, it is observed as a triplet.

The aromaticity of **4b-Cl** was further evaluated by density func-tional theory (DFT) calculations. Initially, the nucleus-independent chemical shift (NICS)[63] was calculated on a simplified **[4b]′**, in which the PPh₃ ligands and the PPh₃⁺ were replaced by PH₃ and PH₃⁺, respectively. The NICS(1)$_{zz}$ values for the three rings were obtained as −20.1, −12.6 and −15.0 ppm, respectively (Fig. 3a). For comparison, the NICS(1)$_{zz}$ of benzene was calculated, and a value of −29.8 ppm indi-cates stronger aromaticity (Supplementary Fig. 95). Subsequently, the anisotropy of the induced current density (ACID)[64] method was applied on **[4b]′**, and a clockwise diatropic ring current passing through the periphery of the entire fused-ring could be obviously observed (Fig. 3b, and several higher resolution pictures are shown in Supplementary Fig. 89). In addition, the aromatic stabilization energy (ASE) using the "methyl-methylene isomerization method"[65] was eval-uated, and the energies of the three reactions (**[4b + Me]** → **[iso −4b + Me]**, **[4b + Me]** → **[iso−4b + Me]′**, and **[4b + Me]** → **[iso −4b + Me]″**) are 14.0, 22.4 and 31.1 kcal/mol, respectively (Fig. 3c). The negative NICS(1)$_{zz}$ values, the clockwise diatropic ring current, and the remarkable positive ASE values, all point to the aromaticity of **4b-Cl** as a whole. Noteworthily, similar calculations on **[3b]′** and **[3b + Me]**, two simplified model complexes of **3b**, were also carried out (Supple-mentary Fig. 90), and the results indicate that osmabenzooxirene-fused osmapentalenes are also aromatic.

## Reactivity studies of metallabenzyne-fused metallapentalenes
As mentioned previously, the carbyne carbon bond angle in complex **4b-Cl** is 146.65(15)°, far from the ideal bond angle towards the sp-hybridized carbon, indicating there is considerable ring strain. DFT calculations were performed to estimate the strain value. When the simplified model **[4b-1]** was used for calculation, the computed energy is −11.2 kcal/mol (Fig. 4a). The results suggested the prepared osmabenzyne-fused osmapentalenes might show versatile reactivities.

downfield position of 11.25 ppm, and the C3H, C5H, C6H and C8H appear at a typical aromatic region of 8.16–6.93 ppm. Notably, the C10H is observed at a relatively upfield position of 3.96 ppm, falling within the range observed for Os≡C-CH protons in classical

Thereafter, complex **4a** was selected as the starting material to test the reactivities. Electrophilic reactions were firstly investigated. When HBF$_4$·Et$_2$O was added to a solution of **4a** in CD$_2$Cl$_2$ at rt, complex **5** showing two $^{31}$P signals at 16.97 and 0.19 ppm was detected. However, this complex is too unstable to be isolated and fully characterized, and it could be converted back to **4a** upon the addition of NaHCO$_3$. We propose that complex **5** is a 16-electron osmabenzene-fused osmapentalene as shown in Fig. 5a, which was supported by its HRMS data (calculated $m/z = 597.1639$, found $m/z = 597.1662$; Supplementary Fig. 68). This reaction shares similarities with the ones of osmapentalynes with HBF$_4$·Et$_2$O that generated 16-electron osmapentalenes[45,48,56], except that no Os≡C shift product was detected in this case. This is not surprising because if the Os≡C bond in **4a** had shifted to the five-membered ring, the carbyne carbon bond angle would be smaller so as to lead to greater ring strain, therefore the resulting osmabenzene-fused osmapentalyne that would be the isomer of **4a** should have higher energy and be more unstable, which was further supported by theoretical results (the hypothetical osmabenzene-fused osmapentalyne **[4a]'** is higher in energy by 11.0 kcal/mol compared to **4a**, Supplementary Fig. 94).

To obtain Os≡C shifted complexes, reactions with another electrophile, TsCl, were attempted. When **4a** was treated with TsCl in CH$_2$Cl$_2$ at rt, complex **6** with an Os≡C in one of its five-membered rings was fortunately isolated in 78% yield (Fig. 5a).

From the solid-state structure exhibited in Fig. 5b (left), complex **6** still contains a planar [5,5,6]-fused tricyclic configuration (0.027 Å as the mean deviation of the least-squares plane consisting Os1 and C1-C11). The C–C bond lengths within the fused system are between 1.353(4) and 1.452(4) Å, comparable to those of **4a** (1.356(3)-1.450(2) Å). The Os1–C1 distance is 1.848(3) Å, similar to that of Os1–C11 in complex **4a** (1.8503(18) Å), indicating the Os≡C bond in the six-membered ring of **4a** shifted to the five-membered ring during the reaction process, leading to the formation of an osmabenzene-fused osmapentalyne. The carbyne carbon bond angle (Os1–C1–C2) is 130.5(2)°, within the range of those in the previously reported osmapentalynes (127.9-131.2°)[48,55]. The significantly smaller bond angle compared to that of Os1–C11–C10 (146.65(15)°) in complex **4a** demonstrated greater ring strain, just as was mentioned previously. To further estimate the ring strain, DFT calculations on the simplified model **[6-1]** were performed,

**Fig. 4 | Calculated strain Energies. a** Calculated strain energy of **[4b-1]**. **b** Calculated strain energy of **[6-1]**.

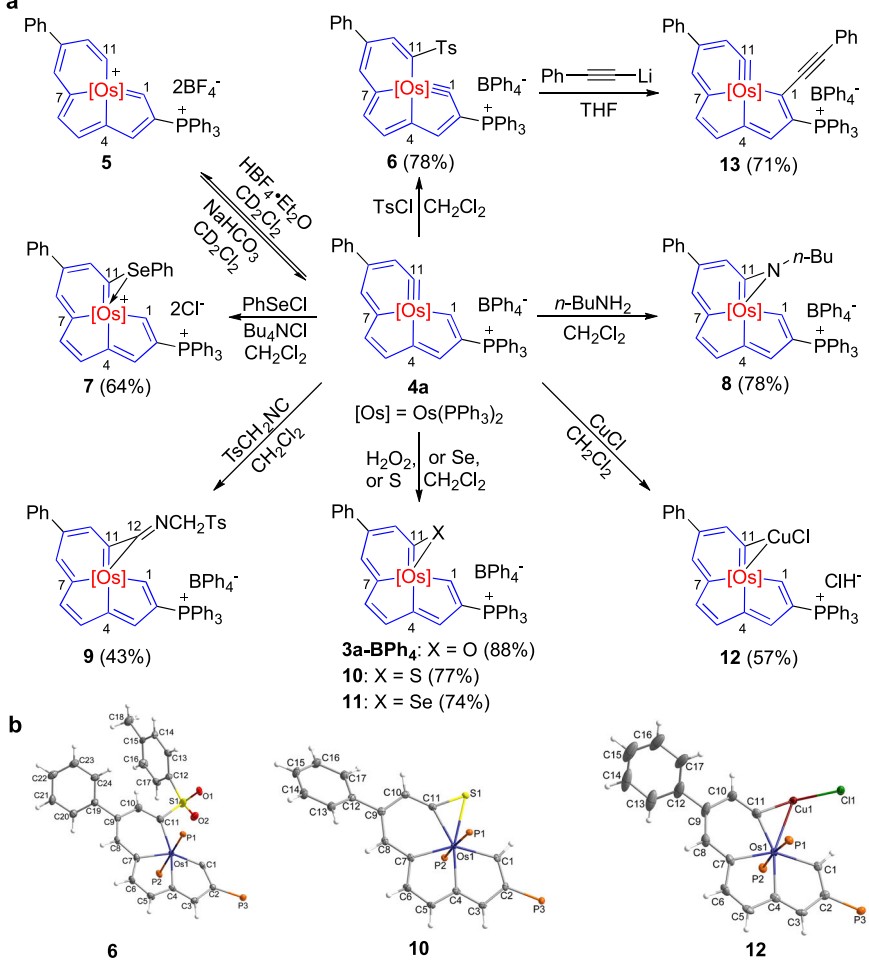

**Fig. 5 | Reactivities of 4a and 6 and the structures of the products. a** Reactivities of **4a** and **6**. **b** Single-crystal X-ray structures for the cations of complexes **6** (left), **10** (middle), and **12** (right) with thermal ellipsoids drawn at the 50% probability level (phenyl groups in PPh$_3$ have been omitted for clarity).

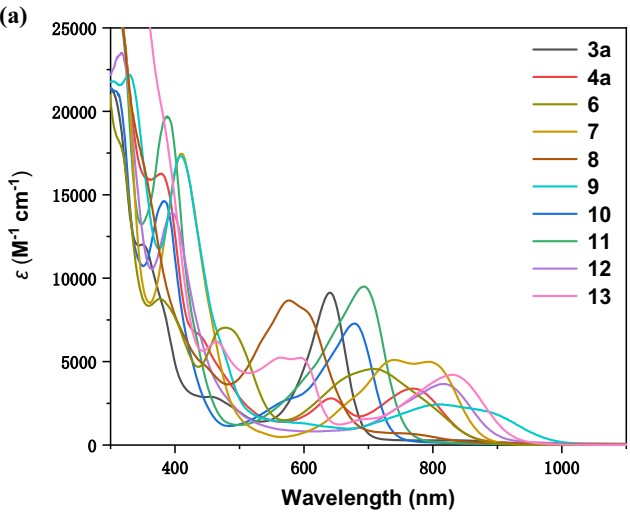

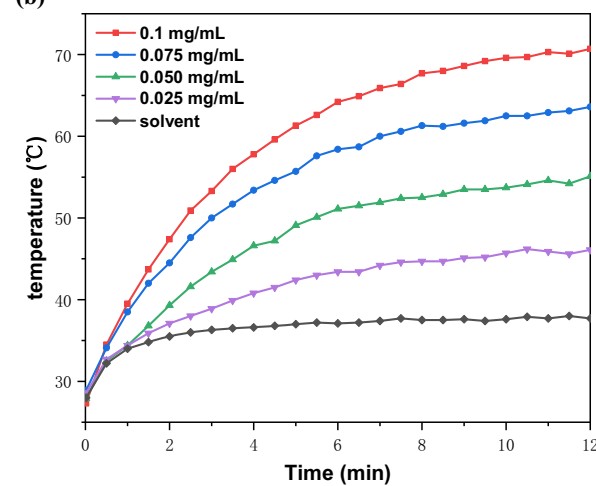

**Fig. 6 | UV-Vis-NIR absorptions and photothermal property. a** UV-Vis-NIR absorption spectra of **3a, 4a**, and **6-13** measured in CH$_2$Cl$_2$ (1×10$^{-4}$ mol/L) at rt. **b** Photothermal conversion of **7** at different concentrations (0.025–0.1 mg/mL,

1 mL) and in DMSO solvent (1 mL) under 808 nm (1 W/cm$^2$) laser irradiation (area of irradiation: 1 cm$^2$) for 12 mins.

and a value of −19.9 kcal/mol was obtained (Fig. 4b). Therefore, this M≡C shift is from a ring with lower strain to a ring with higher strain, which is different from those within the metallapentalyne rings[45,48,56]. Although complex **6** has higher ring strain, further theoretical results indicate that the Gibbs free energy (ΔG) of the reaction is -1.9 kcal/mol (Supplementary Fig. 92). Besides, **6** is lower in energy by 13.4 kcal/mol in comparison with its hypothetical isomer (**[6]'**) where the Ts group is located at the six-membered ring, which is probably because the two large groups, Ts and PPh$_3^+$, are spatially repulsive in **[6]'** (Supplementary Fig. 93). Of note, the C11 signal in the $^{13}$C{$^1$H} NMR spectrum of **6** is at 323.41 ppm, close to that of the Os≡C atom of **4b-Cl** (334.77 ppm). The difference is that this signal is a triplet of doublets (td), attributed to the coupling of the two Os-*P*Ph$_3$ ligands together with the PPh$_3^+$ (the Os≡C signal **4b-Cl** is a triplet, as mentioned previously).

The electrophile PhSeCl is also reactive with **4a**. In the presence of Bu$_4$NCl, **4a** and PhSeCl were converted into complex **7**, whose X-ray single crystal structure is displayed in Supplementary Fig. 84, in 64% yield (Fig. 5a).

Interestingly, **4a** is also reactive with nucleophiles. For example, when **4a** was handled with *n*-BuNH$_2$ in air, complex **8** was obtained in 78% yield via nucleophilic attack of the NH$_2$ group to the Os≡C bond (Fig. 5a)[48]. TsCH$_2$NC was tested as the other nucleophile, and complex **9** was isolated in 43% yield (Fig. 5a). It is worth noting that complex **9** is a 12C-carbolong complex in which all the twelve carbon atoms in the backbone are *sp*$^2$-hybridized[53]. The results indicate the metal−carbyne bond in complex **4a** is ambiphilic, similar to that of metallapentalynes[3,66,67].

Oxidation reactions are applicable for **4a**. Treatment of **4a** with H$_2$O$_2$ afforded complex **3a-BPh$_4$** in 88% yield (Fig. 5a). The cation of **3a-BPh$_4$** is exactly the same as that of **3a**, and the only difference is the anion. S$_8$ and Se are also suitable oxidants, and when they were treated with **4a**, complexes **10** and **11** generated yields of 77% and 74%, respectively (Fig. 5a). The only difference between **10, 11** and **3a-BPh$_4$** is the heteroatoms present on their three-membered rings (S for **10**, Se for **11**, and O for **3a-BPh$_4$**). These complexes were fully characterized by NMR spectroscopy and HRMS spectrometry. In addition, the structures of complexes **10** (Fig. 5b, middle) and **11** (Supplementary Fig. 86) were further confirmed by X-ray single crystal diffraction technique.

The reactivity of complex **4a** towards CuCl was next carried out, and the dimetallic carbolong complex **12** was formed in 57% yield

(Fig. 5a). Noteworthily, such kind of reactions has been found for metallapentalynes[68], but not for metallabenzynes. The solid-state structure of **12** is displayed in Fig. 5b (right). The four fused rings are still almost planar, which is reflected by the mean deviation of 0.033 Å from the least squares plane consisting Os1, C1−C11, and Cu1, and the $^{31}$P singlet at -6.37 ppm for the two Os-*P*Ph$_3$ ligands. It should be noted that the BPh$_4^-$ anion in **4a** was exchanged by Cl$^-$ from excess of CuCl during the reaction.

In consideration that complex **6** also contains a highly strained ring, we further briefly investigated its reactivity. When PhC≡CLi was added to a THF solution of **4a** at -10 °C, the nucleophilic alkynyl anion attacked the carbyne carbon of **4a** with accompanying loss of the Ts group, and complex **13** was generated in 71% yield (Fig. 5a). Complex **13** has been characterized by NMR spectroscopy and HRMS spectrometry. The C11 atom resonates at 330.19 ppm as a triplet in the $^{13}$C{$^1$H} NMR spectrum, which is similar to the Os≡C atom of **4b-Cl** and indicates the Os≡C bond shifted back to the six-membered ring (note: if the Os≡C bond is in the five-membered ring, its Os≡C signal would exhibit as a triplet of doublets, like that of **6** as discussed previously).

## Photophysical and photothermal properties

The UV-Vis-NIR absorption spectra were collected for complexes **3a, 4a**, and **6-13**. As shown in Fig. 6a, all these complexes exhibit broad absorption bands in the UV-Vis−NIR region. In the range of 550 to 1000 nm, the absorption maximum of osmabenzyne-fused osmapentalene **4a**, is located at 770 nm (log ε = 3.53). For O-, S-, Se-containing derivatives **3a, 10, 11**, and osmabenzyne-fused osmapentalene **6**, the absorption maxima are blue-shifted to 640 (log ε = 3.96), 678 (log ε = 3.86), 695 (log ε = 3.98), and 710 nm (log ε = 3.66), respectively. The most blue-shifted complex is **8**, with the absorption maximum at 577 nm (log ε = 3.93). On the other hand, the dimetallic complex **12**, the electrophilic addition product **7**, the alkynyl-substituted complex **13**, and the 12C-carbolong complex **9**, are all red-shifted in comparison with complex **4a**, with the absorption maxima at 817 (log ε = 3.56), 793 (log ε = 3.70), 833 (log ε = 3.62) and 810 nm (log ε = 3.39), respectively. Specifically, complex **9** exhibits a remarkably broad absorption that extends to over 1000 nm. These findings indicate that through the reactions of complex **4a** with different reagents, the absorptions of the products can be easily adjusted.

Complex **7**, which has the strongest absorption at 808 nm among these substances, was then chosen to study its photothermal property

under 808 nm laser irradiation (Fig. 6b). Compared to solvent-only conditions, the addition of different amounts of **7** could increase the temperature. When the concentration was 0.1 mg/mL, the temperature rose from rt to about 70 °C within 12 minutes. Furthermore, photo-thermal stability experiments indicated that within five cycles, no obvious degradation emerged (Supplementary Fig. 96).

## Discussion

In summary, we synthesized a series of strained osmabenzyne-fused osmapentalenes, and studied their chemical and physical properties. These complexes are 11C-carbolong complexes with a planar carbon-chain ligand. Due to the existence of an Os≡C bond, the osmabenzyne-fused osmapentalenes show versatile reactivities, including Os≡C shift, electrophilic addition, nucleophilic addition, oxidation, and metala-tion reaction, affording a series of 11C-carbolong derivatives and one 12C-carbolong complex. It is particularly pointed out that the M≡C shift from a 6-membered to a 5-membered ring achieved a previously unknown metal carbyne bond transfer mode, affording the other type of planar 11C-carbolong skeleton, osmabenzene-fused osmapentalyne, which is also a skeleton that combines metallapentalyne and metalla-benzene. These complexes exhibit broad UV-Vis-NIR absorptions and good photothermal stability, which may be potentially applied as photofunctional materials. This work introduces metallabenzyne into carbolong complexes, which further enriches metallaaromatic chemistry.

## Methods

### General procedure for the synthesis of metallabenzooxirene-fused metallapentalenes

To a solution of complex **1** (200 mg, 0.17 mmol) and AgBF$_4$ (102 mg, 0.52 mmol) in dichloromethane (5 mL) was added the corresponding propargylic alcohol (**2a-e**, 0.87 mmol) or propargyl phenyl ether (**2 f**, 0.87 mmol). The reaction mixture was stirred at room temperature for 24 hours. Then the solution was evaporated under vacuum to a volume of ~2 mL. The residue was purified by column chromatography to afford the target metallabenzooxirene-fused metallapentalenes.

### General procedure for the synthesis of metallabenzyne-fused metallapentalenes

Under Ar atmosphere, **3a** and LiAlH$_4$ (4 equiv.) were added step by step to a mixed solution (5 mL) of dichloromethane and ethyl ether (v/v = 1:1). The reaction mixture was stirred at room temperature for 5 mins, filtered under an argon atmosphere and the filtrate was eva-porated in vacuo. Then the residue was dissolved in methanol followed by the addition of a methanol solution of NaBPh$_4$ (2.6 equiv.). After a few seconds, the mixture was filtered under Ar, and the precipitate was washed with methanol. The solid was dissolved in dichloromethane, then *n*-hexane was added to the solution. The precipitate was collected by filtration, washed with *n*-hexane, and dried under vacuum to afford the target metallabenzyne-fused metallapentalenes.

## Data availability

All data relating to the full experimental procedures, spectral data for new complexes, crystallographic details, computational details, and Cartesian coordinates are provided in the Supplementary Information/Source Data file. The data of the X-ray crystallographic structures of complexes **3a**, **3c**, **4b-Cl**, **5**, **6** and **9-11** have been deposited in the Cambridge Crystallographic Data Center under accession numbers CCDC: 2244936 (**3a**), 2244956 (**3c**), 2244958 (**4b-Cl**), 2260544 (**5**), 2244973 (**6**), 2251588 (**9**), 2244957 (**10**), and 2244972 (**11**). The X-ray crystallographic data are available free of charge from The Cambridge Crystallographic Data Center via http:// www.ccdc.cam.ac.uk/structures/. All data are available from the corresponding author upon request. Source data are provided in this paper.

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

## Acknowledgements

This work was supported by the National Natural Science Foundation of China (Nos. 21931002, 22371111, 22071098 and 22101123), the Guangdong Provincial Key Laboratory of Catalysis (No. 2020B121201002), Introduction of Major Talent Projects in Guangdong Province (No. 2019CX01C079), Guangdong Grants (No. 2021ZT09C064), and Outstanding Talents Training Fund in Shenzhen.

## Author contributions

D.C and H.X. devised the project. Y.C., D.C., and H.X. supervised the experimental study. B.X. and W.M. performed the experimental work. Z.L. performed the computational work. B.X., D.C., and H.X. wrote the paper and prepared the supplemental information with input from all authors. All authors discussed the results in detail and commented on the manuscript.

## Competing interests

The authors declare no competing interests.
