## [Peer Review File · Nature Communications]

Syntheses and Reactivities of Strained Fused-Ring Metallaaromatics Containing Planar Eleven-Carbon ChainsReviewers' Comments:

Reviewer #1:

Remarks to the Author:

This paper describes the syntheses and reactions of new fused ring metallaaromatic compounds that have an osmabenzynes ring fused to an osmapentalene core. This type of planar, 11 carbon atom fused-ring metallaaromatic structure is unique and expands the set of related planar carbonyl complexes that have already been reported with 7-10 and 12 carbon atoms. The 11 carbon atom planar fused-ring compounds have been shown convincingly to be aromatic. Regarding reactivity, the most interesting transformation of the new fused-ring compound 4a is that in which the Os-C triple bond moves from the six-membered osmabenzynes ring to one of the five-membered osmapentalene rings upon addition of a tosyl group to the carbyne carbon. This reaction is favorable even though it results in increased ring strain as determined by DFT calculations. The other reactions explored essentially result in formation of additional fused three-membered osmacyclic rings as an atom of the introduced reagent bridges the Os-C triple bond. The new carbonyl compounds exhibit broad UV-vis-NIR absorptions and have relatively high photothermal stability. All the new compounds are well characterized and the experimental work appears to have been carried out carefully and with attention to detail.

In view of the unique nature of the new compounds and their interesting reactions I support publication after the following main points have been addressed.

1. The proposed mechanism shown in SI Fig 1 appears reasonable for the reactions with propargylic alcohols, but it does not explain how or when the phenyl group is lost in the case of the reaction between 1 and 2f to form 3f. How and at what point is the phenyl group lost?
2. In Fig 4, what is DE?
3. The reaction between 4a and tosyl chloride to give 5, which involves a shift of the Os to C triple bond from the six-membered ring to one of the five-membered rings, is very intriguing. Calculations show that this transformation increases the ring strain significantly, but that the overall energy of the product 5 is (as would be expected) lower. The factors (i.e. driving forces) that make this movement of the triple bond thermodynamically favorable need to be considered more closely and discussed. For example, is the ASE of 5 greater than for 4a, etc?
4. If 5 is thermodynamically more stable than 4a, do calculations show the proton analogue of 5 (i.e. H instead of Ts) is also more stable? Does reaction of 4a with strong protic acid (another electrophile) result in rearrangement to the proton analogue of 5?
5. A key observation concerning the fused rings in the 11C carbonyl complexes 5-11 is that they are all co-planar. This is in stark contrast to the reported analogous 11C carbonyl complex reported in ref 55 where the six-membered ring is nearly perpendicular to the plane of the other fused rings. Why are the ring conformations of these compounds so different? Are the thermodynamic isomers isolated in each case? Do the fused 3-membered rings make an important difference? What are the differences of the ASEs? Do the different ring substituents play an important role?

Minor points for consideration:

1. Title: Syntheses and....
2. Title: ...Fused-Ring Metallaaromatics....?
3. Lines 54 and 58, unusual word usage for "regret"
4. Line 82 "satisfactory"?
5. Line 145 "is located"
6. Line 213 "complex in which all"
7. Line 235 "collected for complexes"
8. Sentence on lines 236-238 needs rewording
9. Caption for Fig 6: "and in DMSO solvent"
10. Caption for Fig 6: the solvent volume and area of irradiation should be defined
11. Line 266: the osmaaromatic compound is metalated (with Cu) to form a dimetallic product. It does

not undergo demetallation.

12. Line 287 "...mins, filtered under an argon atmosphere...."

Reviewer #2:

Remarks to the Author:

This manuscript describes the synthesis and properties of C-11 carbonyl complexes, which have never been reported previously. The reactivity of a C-11 carbonyl complex toward various elements was thoroughly investigated. The photochemical properties of the products were also measured, revealing a broad scope of absorption depending on the element used in the reaction. The synthesis, reactivity, and aromaticity of the product were comprehensively explored, making it of great interest to chemists working with metallaaromatics. The synthetic method used to obtain compound 4 is particularly intriguing; it presents a novel approach to obtain a metal-carbon triple bond.

Thus, I believe the work will be highly appreciated by the researchers working on metallaaromatic compounds. Unfortunately, however, I have feeling that the content lacks generality to appeal to the readers of Nature communication and is better fit to specialized journals for chemistry.

As more specific comment, I found the triple bond shift from 6-membered ring to 5-membered ring is also very interesting and wonder whether 5 in Figure 5 is just a meta-stable kinetic product or the more stable isomer than its 6-membered ring one. It would be interesting if the Ts-group at the 11-position is replaced by a hydrogen by hydride reduction. I understand it is difficult, but at least, additional calculations on Ts- (or mesyl) substituted 4b-1 and 5-1 would be helpful.

Reviewer #3:

Remarks to the Author:

From an experimental viewpoint, the results reported in this manuscript are novel and interesting, especially the observation of shift of the metal-carbon "triple" bond from a 6-membered to a 5-membered ring, but the theoretical side needs further work.

The authors draw structures with Os-C "single", "double" and "triple" bonds. Can this be justified using more convincing arguments rather than simple electron-counting rules? For example, the lengths of the Os-C bonds, three "single" and one "triple", in 4b-Cl' (measured from the Cartesian coordinates reported in the SI) are 2.10, 2.14, 2.10 and 1.86 Å, respectively, which is not that much of a difference for metal-carbon bonds.

The NICS(1)zz values shown in Fig 3a suggest that ring A is the more aromatic than rings B and C. However, the bond alternation between the C-C bonds in this ring is more pronounced than those in rings B and C and therefore, ring A should be less aromatic than B and C. The presence of the two PH3 groups attached to the Os atom and placed above and below the near-planar arrangement of rings A, B and C influence the NICS(1)zz values to a significant extent and, as a result, in this situation NICS(1)zz becomes an unreliable measure of aromaticity. It would be more appropriate to calculate and discuss NICS(0) and NICS(0)zz values instead.

The ACID plot shown in Fig 3b appears to show mainly a peripheral ring current rather than currents within the individual rings A, B and C. The level of detail makes it difficult to see the extent to which that peripheral ring current passes through the Os atom and, in any case, the presence of the PH3 groups attached to this atom would make a meaningful analysis of the ACID plot far from straightforward. It would be good to see more detailed ACID plots (at least, in the SI), and a more thorough analysis of these plots.

The method used by the authors for geometry optimizations, B3LYP/6-31G* (with SDD on Os), does not include dispersion corrections and the basis set is of a very modest size. I would suggest re-

doing all geometry optimizations at the B3LYP-D3(BJ)/def2-TZVP level which was used for single-point calculations. The lowest vibrational frequencies should be included together with the optimized geometries in the SI because provide information about the rigidity of these geometries. The authors have reported extensive crystallographic data and these should be compared to the optimized geometries.

p 6, lines 156-157: The three NICS(1)zz values should have been compared to the NICS(1)zz value for benzene calculated at the same level of theory.

p 7, line 176: The authors state that "DFT calculations were performed to estimate the strain value" but no such values have been reported.

Fig 4a and b show cationic reactants but no cationic products. This should be looked at and corrected.

Thank you very much for handling our manuscript. We also thank the reviewers for their insightful comments. All the points raised by the reviewers have been carefully considered and the changes have been highlighted in the revised manuscript. Now the manuscript has been revised accordingly.

Reviewer 1:

Comments:

This paper describes the syntheses and reactions of new fused ring metallaaromatic compounds that have an osmabenzene ring fused to an osmapentalene core. This type of planar, 11 carbon atom fused-ring metallaaromatic structure is unique and expands the set of related planar carbonyl complexes that have already been reported with 7-10 and 12 carbon atoms. The 11 carbon atom planar fused-ring compounds have been shown convincingly to be aromatic. Regarding reactivity, the most interesting transformation of the new fused-ring compound **4a** is that in which the Os-C triple bond moves from the six-membered osmabenzene ring to one of the five-membered osmapentalene rings upon addition of a tosyl group to the carbyne carbon. This reaction is favorable even though it results in increased ring strain as determined by DFT calculations. The other reactions explored essentially result in formation of additional fused three-membered osmacyclic rings as an atom of the introduced reagent bridges the Os-C triple bond. The new carbonyl compounds exhibit broad UV-vis-NIR absorptions and have relatively high photothermal stability. All the new compounds are well characterized and the experimental work appears to have been carried out carefully and with attention to detail.

In view of the unique nature of the new compounds and their interesting reactions I support publication after the following main points have been addressed.

Response: Thanks very much for your positive comments.

1. The proposed mechanism shown in SI Fig. 1 appears reasonable for the reactions with propargylic alcohols, but it does not explain how or when the phenyl group is lost in the case of the reaction between **1** and **2f** to form **3f**. How and at what point is the phenyl group lost?

Response: Thanks very much for your comment. We are sorry that we did not consider how the phenyl group was lost in the case of the reaction between **1** and **2f**. As suggested, the reaction of propargyl phenyl ether (**2f**) with **1** and AgBF₄ in the presence of excess H₂¹⁸O was conducted, and we found that the product was also labeled by ¹⁸O (**¹⁸O-3f**) in the yield of 29% (Supplementary Fig. 1), as evidenced by HRMS (Supplementary Fig. 79). It means that it is H₂O from air or solvent that attacked the skeleton of **2f** and caused cleavage of the PhO-C bond during the reaction.

A sentence of "To further determine the situation of propargyl phenyl ether (**2f**), the reaction of this substance with **1** and AgBF₄ in the presence of excess H₂¹⁸O was conducted, and the product was also labeled by ¹⁸O (**¹⁸O-3f**) in the yield of 29% (Supplementary Fig. 1), as evidenced by HRMS (Supplementary Fig. 79). Therefore, it was H₂O from air or solvent that attacked the skeleton of **2f** and caused cleavage of the

PhO–C bond during the reaction, which is not unexpected considering that the Ar–O bond is typically more inert than alkyl–O bond in aryl alkyl ethers." has been added in the revised manuscript.

2. In Fig 4, what is DE?

Response: We are sorry for the mistake. We have revised "DE" to " ΔE ".

3. The reaction between **4a** and tosyl chloride to give **5**, which involves a shift of the Os to C triple bond from the six-membered ring to one of the five-membered rings, is very intriguing. Calculations show that this transformation increases the ring strain significantly, but that the overall energy of the product **5** is (as would be expected) lower. The factors (i.e. driving forces) that make this movement of the triple bond thermodynamically favorable need to be considered more closely and discussed. For example, is the ASE of **5** greater than for **4a**, etc?

Response: Thanks very much for your comment. We are sorry that we did not carefully discuss this phenomenon. In fact, our previous statement "Although complex **5** has higher ring strain, further theoretical results indicate it is lower in energy of 3.8 kcal/mol compared to **4a**" is not rigorous, because **4a** and **5** (in the new version, it is complex **6**) are not isomers, and their energies can not be compared. We can only calculate the Gibbs free energy (ΔG) of the reaction. The statement has been corrected as "Although complex **6** has higher ring strain, further theoretical results indicate that the Gibbs free energy (ΔG) of the reaction is -1.9 kcal/mol (Supplementary Fig. 92)".

Furthermore, in the revised version, the reaction of **4a** with H^+ has been carried out, and no such $Os\equiv C$ shift could be detected. Instead, it resulted in an unstable 16-electron species **5** which could not be isolated (Note, complex **5** in the new version is a new added complex), and it was converted back to **4a** upon the addition of $NaHCO_3$ (the reversible reactions have been added in Fig. 5a, and the related discussion is included in the manuscript). DFT calculations suggest if $Os\equiv C$ shift had occurred, the targeted product osmabenzene-fused osmapentalyne (**4a'**) would be in higher energy of 11.0 kcal/mol. This is in agreement with the ring strain results (Note, although the Gibbs free energy (ΔG) of the reaction of **4a** with HCl to **5** is 4.2 kcal/mol, it is reasonable that **5** can be detected but not be isolated due to the low ΔG).

However, since the transformation from **4a** to **6** is not an isomerization process, it is not incomprehensible for it to occur. Just like the transformations of complexes **3** to complexes **4** shown in Fig. 2, they are also processes of increasing ring strain.

Furthermore, as suggested, we also calculated the ASE values, and the results suggest that the ASE values of **6** are not significantly different from those of **4** in general.

4. If **5** is thermodynamically more stable than **4a**, do calculations show the proton analogue of **5** (i.e. H instead of Ts) is also more stable? Does reaction of **4a** with strong protic acid (another electrophile) result in rearrangement to the proton analogue of **5**?

Response: Thanks again for your comment. As mentioned for question 3, the proton analogue of **6** (labeled as complex **4a'**) is higher in energy of 11.0 kcal/mol than **4a**, and the reaction of **4a** with HBF_4 did not result in rearrangement. Instead, it resulted in an unstable 16-electron species **5**.

5. A key observation concerning the fused rings in the 11C carbonyl complexes **5-11** is that they are all co-planar. This is in stark contrast to the reported analogous 11C carbonyl complex reported in ref. 55 where the six-membered ring is nearly perpendicular to the plane of the other fused rings. Why are the ring conformations of these compounds so different? Are the thermodynamic isomers isolated in each case? Do the fused 3-membered rings make an important difference? What are the differences of the ASEs? Do the different ring substituents play an important role?

Response: Thanks very much for your comment. As suggested, we have tried to heat several 11C-carbonyl complexes, including **4a**, **3a-BPh₄**, **6**, **10** and **11** (these numbers correspond to the substances in the new version), however, no isomerization was observed before their decomposition in all cases. We also synthesized 11C-carbonyl complexes reported in ref. 55 where the six-membered ring is nearly perpendicular to the plane of the other fused rings, and tried to see if their non-planar skeletons can become planar, however, before their isomerization to η^5 species as described in ref. 55, no change has been detected. In our opinion, the main reason for the non-planar configuration of the previous 11C-carbonyl complexes is that their ancillary ligands are one Cl and one PPh_3 , making them as 16-electron substances if one of the double bonds in the carbon chain had not coordinated with the osmium. Perhaps it is the coordination of the double bond with Os that causes the six-membered ring to be an

irregular non-planar structure. In contrast, in this manuscript, the ancillary ligands are two PPh₃, and the Os center is already 18-electron without the need for double bond coordination.

In addition, DFT calculations have been performed to see if the planar 11C-carbolong complexes could be converted to non-planar isomers. The cations of complexes **4a** and **3a-BPh₄** were selected for calculation, and the results show that their non-planar isomers have much higher energies.

We have also tried some reactions of the planar 11C-carbolong complexes in this manuscript (such as **4a**, **3a-BPh₄**, **6**, **10** and **11**) with some Cl⁻ sources (such as Bu₄NCl and NaCl), to see if one of the PPh₃ ligands could be converted into Cl, however, no reaction was detected before their decomposition. Similarly, some reactions of the non-planar 11C-carbolong complexes in ref. 55 with PPh₃ were also tried, to see if the Cl in those complexes could be converted into PPh₃, however, still no reaction was detected before their conversion to η^5 species.

Therefore, the planar 11C-carbolong complexes in this manuscript is not necessarily related to the non-planar 11C-carbolong complexes in ref. 55. The different structures might be caused by their different ancillary ligands. Of course, we cannot rule out the influence of the fused 3-membered rings, as pointed out by this reviewer.

6. Minor points for consideration:

- (1). Title: Syntheses and.....
- (2). Title: ...Fused-Ring Metallaaromatics....?

Response: Thanks very much for your careful reading and suggestions. Both of them have been revised as suggested.

- (3). Lines 54 and 58, unusual word usage for “regret”.

Response: “it is a regret that...” has been revised as “one imperfect aspect is that...”; “the other regret is that...” has been revised as “the other issue that needs to be mentioned is that...”.

- (4). Line 82 “satisfactory”?
- (5). Line 145 “is located”.
- (6). Line 213 “complex in which all”.
- (7). Line 235 “collected for complexes”.

Response: All of them have been corrected as suggested.

- (8). Sentence on lines 236-238 needs rewording.

Response: The sentence “The absorption maximum in low-energy absorption band of the osmabenzynes-fused osmapentalene **4a**, is 770 nm ($\log \epsilon = 3.53$)” has been revised as “In the range of 550 to 1000 nm, the absorption maximum of osmabenzynes-fused osmapentalene **4a**, is located at 770 nm ($\log \epsilon = 3.53$)”.

(9). Caption for Fig 6: “and in DMSO solvent”.

(10). Caption for Fig 6: the solvent volume and area of irradiation should be defined.

Response: The information has been added as suggested.

(11). Line 266: the osmaaromatic compound is metalated (with Cu) to form a dimetallic product. It does not undergo dimetallation.

Response: It is true that dimetallation is not correct, and “dimetallation reaction” has been corrected as “metalation reaction”.

(12). Line 287 “...mins, filtered under an argon atmosphere...”

Response: It has been corrected as suggested.

Reviewer 2:

Comments:

This manuscript describes the synthesis and properties of C-11 carbonyl complexes, which have never been reported previously. The reactivity of a C-11 carbonyl complex toward various elements was thoroughly investigated. The photochemical properties of the products were also measured, revealing a broad scope of absorption depending on the element used in the reaction. The synthesis, reactivity, and aromaticity of the product were comprehensively explored, making it of great interest to chemists working with metallaaromatics. The synthetic method used to obtain compound **4** is particularly intriguing; it presents a novel approach to obtain a metal-carbon triple bond.

Thus, I believe the work will be highly appreciated by the researchers working on metallaaromatic compounds. Unfortunately, however, I have feeling that the content lacks generality to appeal to the readers of *Nature communication* and is better fit to specialized journals for chemistry.

Response: Thanks very much for your comment. Metallaaromatic chemistry is an interdisciplinary subject of organic chemistry, inorganic chemistry and physical chemistry, and it has attracted a number of chemists in different fields. The parent organic analogues of metallaaromatic complexes may be aromatic, nonaromatic, or even antiaromatic, and these unique systems not only enrich the variety of aromatics, but also broaden our understanding and extend the concept of aromaticity, making metallaaromatic chemistry more and more fascinating. This work combines together two primary types of metallaaromatics, metallabenzynes and carbonyl complexes, for the first time, which we believe will attract the attention of all the chemists working on metallaaromatic chemistry.

Furthermore, this manuscript shows interesting carbene reactivities, which is directly related to alkyne metathesis and will also attract the chemists working on that broad area. Therefore, we believe it is fit for *Nature communication*.

As more specific comment, I found the triple bond shift from 6-membered ring to 5-membered ring is also very interesting and wonder whether **5** in Figure 5 is just a meta-stable kinetic product or the more stable isomer than its 6-membered ring one. It would be interesting if the Ts-group at the 11-position is replaced by a hydrogen by hydride reduction. I understand it is difficult, but at least, additional calculations on Ts- (or mesyl) substituted **4b-1** and **5-1** would be helpful.

Response: Thanks very much for your comment. As also mentioned by reviewer 1, we have tried the reaction of **4a** with HBF₄, and only the 16-electron osmabenzene-fused osmapentalene **5** was detected, which could not be further converted to the Os≡C shift product, but can be converted back to **4a** upon the addition of NaHCO₃.

For the Ts-substituted complex (complex **6** in the new version), we have tried to heat it in dichlorobenzene at 110 °C, however, we have not detected any analyzable signals except for partial decomposition.

DFT calculations were further performed to estimate if **6** is the more stable isomer than its 6-membered ring one (**6'**), and the results suggest **6'** is higher in energy of 13.4 kcal/mol than **6**. This is probably because the two large groups, Ts and PPh₃⁺ group, are spatially repulsive. A sentence “Although complex **6** has higher ring strain, further theoretical results indicate that the Gibbs free energy (ΔG) of the reaction is -1.9 kcal/mol (Supplementary Fig. 92). Besides, **6** is lower in energy of 13.4 kcal/mol in comparison with its hypothetical isomer (**[6']**) where the Ts group is located at the six-membered ring, which is probably because the two large groups, Ts and PPh₃⁺ group, are spatially repulsive in **[6']** (Supplementary Fig. 93).” has been included in the revised manuscript.

As suggested, we also tried the reactions of this complex with some hydride sources such as LiAlH₄, NaBH₄, NaBHET₃ and NaH, however, all the reactions were complicated and gave unidentified mixtures.

Delightedly, inspired by this reviewer’s constructive suggestions, we found that another nucleophile, PhC≡CLi, was reactive with complex **6** and generated a new osmabenzene-fused osmapentalene through the Os≡C shift from the five-membered ring to the six-membered ring. This reaction was added into Fig. 5a and corresponding discussion has been included.

Reviewer 3:

Comments:

From an experimental viewpoint, the results reported in this manuscript are novel and interesting, especially the observation of shift of the metal-carbon "triple" bond from a 6-membered to a 5-membered ring, but the theoretical side needs further work.

Response: Thanks very much for your positive comments.

1. The authors draw structures with Os-C "single", "double" and "triple" bonds. Can this be justified using more convincing arguments rather than simple electron-counting rules? For example, the lengths of the Os-C bonds, three "single" and one "triple", in **4b-Cl'** (measured from the Cartesian coordinates reported in the SI) are 2.10, 2.14, 2.10 and 1.86 Å, respectively, which is not that much of a difference for metal-carbon bonds.

Response: Thanks very much for your comment. We totally agree that it is inaccurate that we drew structures just with Os-C "single", "double" and "triple" bonds. In fact, complex **4b-Cl'** has several resonance structures as shown below, which has also been discussed in previous literatures for metallabenzynes (for instance, ref. 60: *Angew. Chem. Int. Ed.* **40**, 1951-1954). In this manuscript, we only show one of the resonance structures, but we should have mentioned it in our previous version. Now the resonances structures have been added in the SI (Supplementary Fig. 3), and a sentence "Similar to osmabenzynes, **4b-Cl** should also have other resonance structures (Supplementary Fig. 3).^{12,60}" has been included in the maintext.

2. The NICS(1)_{zz} values shown in Fig 3a suggest that ring A is the more aromatic than rings B and C. However, the bond alternation between the C-C bonds in this ring is more pronounced than those in rings B and C and therefore, ring A should be less aromatic than B and C. The presence of the two PH₃ groups attached to the Os atom and placed above and below the near-planar arrangement of rings A, B and C influence the NICS(1)_{zz} values to a significant extent and, as a result, in this situation NICS(1)_{zz} becomes an unreliable measure of aromaticity. It would be more appropriate to calculate and discuss NICS(0) and NICS(0)_{zz} values instead.

Response: Thanks very much for your comment. The reason that we choose NICS(1)_{zz} values is because sometimes the proximate σ electrons would cause induced currents and influence significantly the NICS values (Schleyer, P. v. R. *et al. Org. Lett.* **2006**, *8*, 863; Stanger, A. *Eur. J. Org. Chem.* **2020**, 3120.). In contrast, when the probe is 1 Å above the center of the molecular plane, the effect of the σ electrons would be quite small and the values are much more reliable.

In fact, the H and P of PH₃ are not very close to the 1 Å positions above the centers of the rings (the closest distance is farther than 2.0 Å). To investigate the influence of PH₃. We scanned several positions and the calculated NICS(r)_{zz} values are shown below (r represents the distance above the center of the ring). From the picture, we can see

that the slopes of the curves are relatively small when r is around 1 and -1 Å, which suggests they are not significantly affected by σ electrons and PH_3 . In contrast, when $r = 0$ or $r > 2$, the slope is very large, suggesting their influence is significant.

To further confirm the minor impact of PH_3 , we further replaced them by CO ligands, and the NICS values are not influenced significantly.

NICS(1)zz (unit: ppm)

As suggested, we have also calculated NICS(0) and NICS(0)_{zz}, and their values are -4.3, -2.4, -7.3 ppm and -2.8, 7.8, -8.3 ppm for rings A, B and C, respectively (shown in the table below). The NICS(0)_{zz} values suggest the aromaticity of the model complex is not good (especially ring B), which is not consistent with the NMR, structural parameters, ASE and ACID results. Therefore, in our system, the proximate σ electrons might influence significantly the NICS values.

[4b]'	NICS(0)	NICS(0) _{zz}	NICS(1)	NICS(1) _{zz}
Ring A	-4.3	-2.8	-7.7	-20.1
Ring B	-2.4	7.8	-6.9	-12.6
Ring C	-7.3	-8.3	-6.9	-15.0

Because NICS is not the only criterion to evaluate aromaticity, and NICS(1)_{zz} is more

frequently used to evaluate aromaticity for metallaaromatics (for instance, the metallabenzenes in the following two references also contain large auxiliary ligands such as phosphite and NHC, see Nozaki, K. *et al. Angew. Chem. Int. Ed.* **2022**, *61*, e202117096; Lee, D. *et al. J. Am. Chem. Soc.* **2021**, *143*, 7490), together that NICS(1)_{zz} values are better fitted for our system, after comprehensive consideration we decided not replace them by NICS(0) and NICS(0)_{zz}.

3. The ACID plot shown in Fig 3b appears to show mainly a peripheral ring current rather than currents within the individual rings A, B and C. The level of detail makes it difficult to see the extent to which that peripheral ring current passes through the Os atom and, in any case, the presence of the PH₃ groups attached to this atom would make a meaningful analysis of the ACID plot far from straightforward. It would be good to see more detailed ACID plots (at least, in the SI), and a more thorough analysis of these plots.

Response: Thanks very much for your comment. The fused-ring has aromaticity as a whole, like naphthalene (the ACID map is shown below), therefore, this kind of fused-ring complexes show mainly a peripheral ring current. We are sorry that we did not mention it clearly in our previous version. Now we have revised some sentences to make the statement clearer. For instance, the former sentence “and a clockwise diatropic ring current could be obviously observed (Fig. 3b)” has been revised as “and a clockwise diatropic ring current passing through the periphery of the entire fused-ring could be obviously observed (Fig. 3b)”, and “...all point to the aromaticity of **4b-Cl**” has been revised as “...all point to the aromaticity of **4b-Cl** as a whole”.

As suggested, to eliminate the interference of PH_3 groups, we firstly tried to delete them, however, the resulted ACID plot is not reasonable as shown below, because the electronic structure of the Os center has been greatly changed.

Thereafter, we adjusted the angle of view to minimize the obstruction of the Os center by PH_3 , and several clearer and more intuitive images which are shown below have been placed in Supplementary Fig. 89.

4. The method used by the authors for geometry optimizations, B3LYP/6-31G* (with SDD on Os), does not include dispersion corrections and the basis set is of a very modest size. I would suggest re-doing all geometry optimizations at the B3LYP-D3(BJ)/def2-TZVP level which was used for single-point calculations. The lowest vibrational frequencies should be included together with the optimized geometries in the SI because they provide information about the rigidity of these geometries. The authors have reported extensive crystallographic data and these should be compared to the optimized geometries.

Response: Thanks again for your comment. As suggested, we have re-done all geometry optimizations at the B3LYP-D3(BJ)/def2-TZVP level. Furthermore, as suggested, we have also compared the crystallographic data with the optimized geometries of complexes **4b**, **3a** and **6**, and the results shown below indicate the calculated bond distances are quite close to those in the X-ray structures, with mean

relative deviations of 0.33%, 0.62% and 0.55%, respectively. These results have also been included in Supplementary Tables 13-15.

Supplementary Table 13. Comparison of bond lengths in **4b** from the DFT calculated and experimental data.

4b							
Bond Distances (Å)	Os1–C1	Os1–C4	Os1–C7	Os1–C11	C1–C2	C2–C3	C3–C4
Crystallographic data	2.0961	2.1342	2.0809	1.8603	1.368	1.450	1.363
B3LYP-D3(BJ)/def2-TZVP	2.108	2.1427	2.0976	1.8526	1.371	1.446	1.364
Relative deviations (%)	0.5677	0.3983	0.8025	0.4139	0.2193	0.2759	0.0734
Bond Distances (Å)	C4–C5	C5–C6	C6–C7	C7–C8	C8–C9	C9–C10	C10–C11
Crystallographic data	1.422	1.361	1.439	1.401	1.388	1.424	1.356
B3LYP-D3(BJ)/def2-TZVP	1.418	1.365	1.437	1.403	1.389	1.421	1.346
Relative deviations (%)	0.2813	0.2939	0.1390	0.1428	0.0720	0.2107	0.7375
Mean relative deviations (%)	0.3306						

Supplementary Table 14. Comparison of bond lengths in **3a** from the DFT calculated and experimental data.

3a							
Bond Distances (Å)	Os1–C1	Os1–C4	Os1–C7	Os1–C11	C1–C2	C2–C3	C3–C4
Crystallographic data	2.091	2.079	2.110	2.001	1.373	1.442	1.383
B3LYP-D3(BJ)/def2-TZVP	2.089	2.094	2.135	2.021	1.369	1.44	1.365
Relative deviations (%)	0.0956	0.7215	1.1848	0.9995	0.2913	0.1387	1.3015
Bond Distances (Å)	C4–C5	C5–C6	C6–C7	C7–C8	C8–C9	C9–C10	C10–C11
Crystallographic data	1.414	1.357	1.432	1.393	1.403	1.401	1.393
B3LYP-D3(BJ)/def2-TZVP	1.421	1.354	1.439	1.387	1.405	1.394	1.37
Relative deviations (%)	0.4950	0.2211	0.4888	0.4307	0.1426	0.4996	1.6511
Mean relative deviations (%)	0.6187						

Supplementary Table 15. Comparison of bond lengths in **6** from the DFT calculated and experimental data.

6							
Bond Distances	Os1–C1	Os1–C4	Os1–C7	Os1–C11	C1–C2	C2–C3	C3–C4

(Å)							
Crystallographic data	1.848	2.131	2.071	2.099	1.393	1.428	1.391
B3LYP-D3(BJ)/def2-TZVP	1.844	2.156	2.078	2.104	1.381	1.433	1.383
Relative deviations (%)	0.2164	1.1731	0.3380	0.2382	0.8615	0.3501	0.5751
Bond Distances (Å)	C4–C5	C5–C6	C6–C7	C7–C8	C8–C9	C9–C10	C10–C11
Crystallographic data	1.389	1.377	1.406	1.436	1.359	1.452	1.353
B3LYP-D3(BJ)/def2-TZVP	1.397	1.382	1.409	1.423	1.373	1.446	1.347
Relative deviations (%)	0.5760	0.3631	0.2134	0.9053	1.0301	0.4132	0.4435
Mean relative deviations (%)	0.5498						

5. p6, lines 156-157: The three NICS(1)_{zz} values should have been compared to the NICS(1)_{zz} value for benzene calculated at the same level of theory.

Response: Thanks very much for your comment. As suggested, we have compared the NICS(1)_{zz} value of benzene calculated at the same level of theory, and a sentence “For comparison, the NICS(1)_{zz} of benzene was calculated, and a value of -29.8 indicates stronger aromaticity (Supplementary Fig. 95).” has been added in the manuscript.

6. p7, line 176: The authors state that "DFT calculations were performed to estimate the strain value" but no such values have been reported.

Response: Thanks very much for your comment. It is true that the method in this manuscript to estimate the ring strain is not the same as that reported in the literature. Therefore, we decided to delete that sentence as suggested.

7. Fig 4a and b show cationic reactants but no cationic products. This should be looked at and corrected.

Response: Thanks very much for your careful reading, and the mistake has been corrected.

Thank you very much for your kind consideration. Please let us know of further concerns that arise.

Yours sincerely,
Haiping XIA

Reviewers' Comments:

Reviewer #1:

Remarks to the Author:

[Note from the Editor: Reviewer #1 was asked to look also over the response given to reviewer #2 who was unable to look on the revision again.]

The authors have satisfactorily addressed (and in most cases in considerable depth) all the points raised by Reviewers #1 and #2. The changes made have resulted in an improved manuscript that is now suitable for publication once Reviewer #3 has confirmed the technical issues they raised have also been attended to.

I noted that the language in the newly introduced text of the revised manuscript could be improved in a few instances. The Authors and Editor might want to consider the following suggestions:

page 6 line 14 from end: "...ligands, it is observed as a...."

8 4 : "...resulting osmabenzene-fused..."

8 2 : "...energy by...."

9 line 1: "To obtain Os≡C shifted complexes, reactions with another electrophile, TsCl, were attempted."

10 line 3: "...just as was..."

10 10: "...energy by..."

10 12: "...and PPh₃⁺, are..."

10 13: "...signal in the.....spectrum of 6..."

11 14: "...added to..."

11 15: "...the nucleophilic alkynyl anion attacked the carbyne carbon of 4a with accompanying loss of the Ts group, and complex 13 was generated in 71%....."

11 18: "...triplet in the..."

Reviewer #3:

Remarks to the Author:

The authors have made an effort to respond to all comments of the reviewers and the associated revisions have certainly improved the manuscript. However, there are still details that require further attention (please, see below). In addition, I am not entirely convinced that the manuscript would not be better suited to a more specialized journal: Aromaticity and antiaromaticity which have been emphasized upon in the authors' reply to Reviewer 2 are not the most significant features of the molecules studied in the manuscript; none of these molecules are antiaromatic and the levels of aromaticity suggested by the reported NICS values are significantly lower than that of benzene.

I am not entirely happy with the explanation: "Therefore, it was H₂O from air or solvent that attacked the skeleton of 2f and caused cleavage of the PhO–C bond during the reaction..." The additional experiment described in the same paragraph was carried out in excess H₂(18O), would just the humidity of the lab or the supposedly very small water content in the solvent be sufficient to produce a similar effect?

The lowest vibrational frequency of benzene cannot be 12.79 cm⁻¹ as shown in the the SI; at this level of theory it should be around 412 cm⁻¹ (E_{2u}). This suggests that all other lowest vibrational frequencies should be re-checked, as well.

A couple of minor points:

The 1st sentence of the Introduction reads as if it is incomplete, I would suggest extending it as

"Aromatic compounds are one of the most important species in chemistry".

"one imperfect aspect is ... have never been synthesized" could be replaced by "one notable omission is ... have not been synthesized so far".

Thank you very much for handling our manuscript. We also thank the reviewers for their insightful comments. All the points raised by the reviewers have been carefully considered and the changes have been highlighted in the revised manuscript. Now the manuscript has been revised accordingly.

Reviewer 1:

Comments:

The authors have satisfactorily addressed (and in most cases in considerable depth) all the points raised by Reviewers #1 and #2. The changes made have resulted in an improved manuscript that is now suitable for publication once Reviewer #3 has confirmed the technical issues they raised have also been attended to.

1. I noted that the language in the newly introduced text of the revised manuscript could be improved in a few instances. The Authors and Editor might want to consider the following suggestions:

page 6, line 14, from end: "...ligands, it is observed as a..."

page 8, line 4: "...resulting osmabenzene-fused..."

page 8, line 2: "...energy by..."

page 9, line 1: "To obtain Os≡C shifted complexes, reactions with another electrophile, TsCl, were attempted."

Page 10, line 3: "...just as was..."

Page 10, line 10: "...energy by..."

Page 10, line 12: "...and PPh₃⁺, are..."

Page 10, line 13: "...signal in the.....spectrum of 6..."

Page 11, line 14: "...added to..."

Page 11, line 15: "...the nucleophilic alkynyl anion attacked the carbyne carbon of 4a with accompanying loss of the Ts group, and complex 13 was generated in 71%....."

Page 11, line 18: "...triplet in the..."

Response: Thanks very much for your comment. They are important for improving the language, and all of them have been revised accordingly.

Reviewer 3:

Comments:

The authors have made an effort to respond to all comments of the reviewers and the associated revisions have certainly improved the manuscript. However, there are still details that require further attention (please, see below). In addition, I am not entirely convinced that the manuscript would not be better suited to a more specialized journal: Aromaticity and antiaromaticity which have been emphasized upon in the authors' reply to Reviewer 2 are not the most significant features of the molecules studied in the manuscript; none of these molecules are antiaromatic and the levels of aromaticity suggested by the reported NICS values are significantly lower than that of benzene.

1. I am not entirely happy with the explanation: "Therefore, it was H₂O from air or solvent that attacked the skeleton of **2f** and caused cleavage of the PhO–C bond during the reaction..." The additional experiment described in the same paragraph was carried out in excess H₂(¹⁸O), would just the humidity of the lab or the supposedly very small water content in the solvent be sufficient to produce a similar effect?

Response: Thanks very much for your comment. As suggested, we have tried the reaction of complex **1** with propargyl phenyl ether (**2f**) in the presence of AgBF₄ in a glove box filled with N₂ in extra-dry CH₂Cl₂ (the concentration of H₂O is 50 ppm), but besides some unidentified intermediates, only trace of **3f** was detected. We have also tried the same reaction filled with O₂ in extra-dry CH₂Cl₂, still no **3f** was detected. In fact, under our reaction conditions, the H₂O concentration of 0.06% in CH₂Cl₂ was enough to convert all the starting materials into the final product **3f**, because the molecular weight of complex **1** is as high as 1151. The H₂O concentration of the commonly used CH₂Cl₂ in our lab is labeled as 0.05%. Therefore, H₂O from the solvent, together with some moisture from air, is enough for the reaction. Additional water would cause the reaction yield to be a little lower, therefore we did not add extra H₂O to the reaction mixture. A sentence "We also tried the reaction of **2f** with **1** and AgBF₄ in extra-dry CH₂Cl₂ (the concentration of H₂O is 50 ppm) under a N₂ atmosphere, however, only trace of **3f** was detected." has been added in the manuscript.

2. The lowest vibrational frequency of benzene cannot be 12.79 cm⁻¹ as shown in the the SI; at this level of theory it should be around 412 cm⁻¹ (E2u). This suggests that all other lowest vibrational frequencies should be re-checked, as well.

Response: Thanks very much for your careful reading. We are sorry for the mistake. It was because when we were pasting the number 412.79 cm⁻¹, we mistakenly pasted it as the number 12.79 cm⁻¹, and we missed the first digit 4. We have re-checked all other numbers, and they are all correct.

3. A couple of minor points:

(1) The 1st sentence of the Introduction reads as if it is incomplete, I would suggest extending it as "Aromatic compounds are one of the most important species in chemistry".

(2) "one imperfect aspect is ... have never been synthesized" could be replaced by "one notable omission is ... have not been synthesized so far".

Response: Thanks very much for your comment. They are important for improving the language, and both of them have been revised accordingly.

By the way, we have uploaded two different types of .pdf figures with different sizes, together with one chemdraw type, because we are not sure which type better meets the requirements of the Journal.

Thank you very much for your kind consideration. Please let us know of further

concerns that arise.

Yours sincerely,
Haiping XIA